# Relationships between markers of emotional and social cognition and acoustic-verbal hallucinations in children and adolescents with post-traumatic stress disorder (PTSD)

**Louise-Emilie Dumas**[1,2,3], **Florence Askenazy**[1,2,3‡], **Arnaud Fernandez**[1,2,3‡*]

1 Suicidology and Consultation-Liaison Psychiatry Unit, University Department of Child and Adolescent Psychiatry, University Pediatric Hospital–Lenval, Nice, France, 2 Université Côte d'Azur, Cognition Behaviour Technology (CoBTeK) Laboratory, Nice Cedex, France, 3 Regional Center for Psychotrauma (CRP), Nice—PACA, Corsica, France

‡ These authors are co-senior authors on this work.
* Arnaud.fernandez@hpu.lenval.com

## Abstract

### Introduction

Acoustic-verbal hallucinations (AVH) occur in children and adolescents without psychotic disorders. They are often associated with anxiety, thymic and behavioral disorders and a history of trauma, notably post-traumatic stress disorder (PTSD). AVH may be transient, but their persistence increases the risk of progression to a psychotic disorder. The aim of this study was to observe the links between markers of emotional and social cognition and the presence of AVH in children with PTSD, as well as the evolution of post-traumatic and psychotic symptoms.

### Methods

This was a prospective 6-month study, including children aged 8–16 with PTSD and without psychotic disorder (DSM-5). Participants were divided into two groups, with and without AVH. Emotional cognition markers were measured using the DES IV and BAVQ-R, while social cognition markers were assessed via the NEPSY II test.

### Results

31 patients were included: 16 with AVH and 15 without. Results showed that at inclusion, markers of emotional and social cognition were not associated with AVH. At 6 months, markers of emotional cognition were significantly associated with the persistence of AVH, PTSD and psychotic disorders, unlike those of social cognition.

**Data availability statement:** All relevant data are within the manuscript and its Supporting information files.

**Funding:** The author(s) received no specific funding for this work.

**Competing interests:** The authors have declared that no competing interests exist.

## Conclusion

Emotional cognition markers play a central role in the evolution of hallucinatory, post-traumatic and psychotic symptoms, and could become a target for prevention and targeted therapy.

## Clinical trial registration

ClinicalTrials.gov NCT03356028

## Introduction

Hallucinatory experiences are a symptomatology present in the clinical population apart from a psychotic disorder, organic etiology (metabolic, genetic, infectious, neurological or toxicological) or physiological etiology (such as hypnagogic or hypnopompic hallucinations) [1,2]. These hallucinations are more particularly described in the pediatric population with a prevalence ranging from 17% in 9–12 year olds and 7.5% in 13–18 year olds [3]. They can be multisensory, although acoustic-verbal hallucinations (AVH) are the most reported with a prevalence of 12% in children and adolescents [4–7]. They are linked to the developmental phenomenon of psychic immaturity [8–10] and most often considered transient and benign [8,11].

However, the hallucinatory experience of children and adolescents can be described as associated with thymic, anxiety and behavioral disorders [12,13], a high suicidal risk [14,15] and represent a risk of impairment in the child's day-to-day functioning [11]. The persistence of hallucinations into adolescence represents a poor psychiatric prognostic factor: hallucinations are more congruent with the disorder with which they are associated [10,16,17] and may progressively evolve into a schizophrenia spectrum disorder [9,18,19].

The literature highlights links between exposure to early trauma and the development of hallucinatory experience [20,21]. AVH in the general population is thought to be closely linked to childhood trauma [22,23], particularly in situations of abuse (physical, psychological and sexual abuse) and neglect [22,24]. Studies show that repeated traumatic events increase the risk of PTSD [25] as well as other psychiatric comorbidities [26] with, in particular, hallucinatory experiences and a risk of progression to a psychotic disorder [27–31]. This clinical evolution is described according to a "continuum" model starting from the traumatic event, the hallucinatory experience and up to the development of psychotic disorder [32].

AVH exists in the pediatric population and may be potentially prodromal to a risk of progression to a psychotic disorder if it is associated with a traumatic context [20,21] and persists over time [9,18,19]. We believe it is necessary to understand the mechanisms at play in this evolution in order to propose targeted preventive management.

The scientific literature reports various cognitive impairments described in PTSD as well as in prodromal hallucinatory experiences of a psychotic disorder [33,34]. PTSD symptoms have been linked to dysfunctions in brain structures involved in the sensory and emotional modulation system [35–37]. Sensory dysregulation may result

in hyperreactivity (hypervigilance, flashbacks, freeze response) [38] or hyporeactivity (dissociation, depersonalization, derealization) [39,40]. Emotional and social cognition refers to the set of cognitive processes involved in the recognition, interpretation, and regulation of emotions, as well as in the understanding of others' mental states (intentions, beliefs, affects), which are essential for appropriate social adaptation [41]. In PTSD, impairments in emotional and social cognition lead to the misattribution of threat to neutral stimuli [42] and to biases in the recognition of others' intentions and facial expressions [43,44], resulting in developmental and psychosocial difficulties in children and adolescents [45]. Several studies highlight the role of emotional and social cognition in the emergence of psychotic symptoms and in the progression toward schizophrenia spectrum disorders [46,47]. Emotional dysregulation [48,49], along with deficits in theory of mind and affect recognition [1,50], have been proposed as underlying mechanisms of the reality distortion that characterizes hallucinatory experiences [51].

An initial pilot study was carried out by the team from the University Department of Child and Adolescent Psychiatry of the University Pediatric Hospital-Lenval in Nice [52], which found a significant link between the persistence of AVH at 6 months and PTSD (p<0.05). The study presented here was replicated using the same methodology in a pediatric population with PTSD [53].

In this study we support the hypothesis that markers of emotional and social cognition are involved in the presence and persistence of AVH in children and adolescents with PTSD, as well as during post-traumatic and psychotic symptoms.

The primary objective of our study is to compare emotional and social cognition markers in a population of children and adolescents with PTSD, between those presenting with AVH and those without AVH.

Specifically, we aim to:

1. Compare emotional and social cognition markers in a population of children and adolescents with PTSD, between those presenting with persistent auditory AVH at 6-month follow-up and those without AVH at 6 months;

2. Assess the correlations between emotional and social cognition markers and the presence of AVH at baseline and their persistence at 6-month follow-up;

3. Examine the associations between emotional and social cognition markers and the clinical evolution of PTSD between baseline and 6-month follow-up;

4. Investigate the associations between emotional and social cognition markers and the emergence of a psychotic disorder at 6-month follow-up."

## Methods

### Study design

This study is a prospective, monocentric, case-control, open-label, and longitudinal biomedical observational protocol conducted over two years. It is taking place at the University Department of Child and Adolescent Psychiatry of the University Pediatric Hospital-Lenval in Nice. The PHYSALIS study was launched on February 8, 2020, and is planned to run for a total of seven years, including a five-year recruitment period followed by a two-year follow-up phase. The current results represent a six-month interim analysis.

### Participants

This study is an ancillary study to the protocol entitled "Child and adolescent psychiatry and multidisciplinary research (public health, psychodynamics, neurosciences, and human and social sciences) in children exposed to the Nice attack of July 14, 2016 (Program 14-7)." To facilitate patient recruitment, we requested a revision of the amendment to this ancillary study, allowing the expansion of inclusion criteria to patients with PTSD related to any type of trauma. Participants were

children and adolescents treated at the University Department of Child and Adolescent Psychiatry of the University Pediatric Hospital-Lenval (Nice) for a diagnosis of Post Traumatic Stress Disorder (PTSD) (DSM-5).

**Inclusion criteria:**

• boys and girls aged 8 to 16 years;

• without intellectual deficits (IQ > 70);

• with a diagnosis of PTSD according to DSM-5 criteria (K-SADS-PL).

**Exclusion criteria.**

• a diagnosis of schizophrenic spectrum disorder according to DSM-5 criteria (K-SADS-PL);

• the presence of genetic, neurological, neurodevelopmental (including autism spectrum disorders) and neurosensory pathologies.

This case-control model observes a population with PTSD. Subjects included are divided into two groups: a "case" group, with the presence of AVH, and a "control" group, without AVH. Participants in both groups will be matched for sex and age (+/- 6 months).

## Clinical evaluation tools

**Assessment of inclusion and non-inclusion criteria.** The "Post Traumatic Stress Disorder" (PTSD) and "Psychosis" sections of the psychometric Kiddie Schedule for Affective Disorders and Schizophrenia for School-Age Children-Present and Lifetime (K-SADS-PL) test [54], are conducted in the presence of the child and parents, and used to validate the diagnosis of PTSD and the absence of psychotic disorders. The K-SADS-PL test is a semi-structured diagnostic interview to assess current and past psychopathological episodes in children and adolescents based on DSM-5 criteria, and which is used in a study validating a diagnostic tool for PTSD in a French clinical population [55].

The absence of mental retardation is determined using the abbreviated form of the Wechsler Intelligence Scale for Children (WISC) IV. The French version of the abbreviated form of the WISC IV (Similarities, Picture Concepts, Digit Span, Symbol Search) [56] was administered to screen for intellectual disability (Full Scale IQ < 70) as a non-inclusion criterion; this short form has demonstrated good sensitivity and specificity for screening purposes. The IQ estimate obtained from this abbreviated form was used exclusively for screening and was not included in group comparisons or correlations. All assessments were conducted by the same evaluator trained in test administration by the service's neuropsychologists.

The psychometric test Mini International Neuropsychiatric Interview Enfants-Adolescents (MINI-Kid) [57], validated in French version [58] explores in a standardized way the main psychiatric disorders of the DSM-5 axis in children. The interview is conducted with the child and his or her parents. The presence of an initial psychiatric disorder is categorized using the MINI-Kid: mood disorder: (depression and emotional dysregulation); anxiety disorder (panic disorder, agoraphobia, social phobia, obsessive-compulsive disorder and generalized anxiety); behavior disorder (conduct disorder, oppositional defiant disorder (ODD) and attention-deficit/hyperactivity disorder (ADHD)).

A sociodemographic and clinical questionnaire, used in a previous study [52], collects information on perinatal, medical, surgical, psychological and psychiatric history, psychomotor development, family history, biographical information and family environment. A specific section is devoted to the clinical description of AVH, excluding hypnopompic and hypnagogic episodes, imaginary productions and companions. This clinical description details the hallucinations, analyzing their characteristics for a qualitative assessment of the symptom: age of onset, clinical type, content, frequency, tone of localization and number of voices heard.

**Assessment for group allocation.** The AVH screening questionnaire, used in a previous study [52], classifies participants into "case" or "control" groups. This questionnaire is based on the "Schizophrenia" section of the Diagnostic

Interview Schedule for Children-Child (DISC-C) [59,60], two questions of which deal with the presence of AVH. All participants in the study were asked the following question by the evaluator: "Have you ever heard a voice in your head that is different from your own, and that no one else hears but you?" to determine whether they belonged to the "case" or "control" group.

**Assessment of the emotional cognition marker.** Emotional cognition is assessed using the Differential Emotion Scale IV (DES IV) [61], in its French version [62]. This self-report scale measures 12 subjective emotional experiences: interest, joy, surprise, anger, contempt, disgust, sadness, fear, guilt, shame, shyness and self-hostility. This instrument measures emotional traits representing stable individual differences, expressed by the frequency with which emotions are experienced by the patient daily. The DES-IV has demonstrated adequate construct validity and acceptable internal consistency for assessing discrete emotions in children and adolescents [63,64].

The Beliefs About Voices Questionnaire-Revised (BAVQ-R) [65,66] is a self-assessment scale that characterizes patients' relationship with their hallucinations, validated in the French version [67]. The BAVQ-R has shown good internal consistency and satisfactory psychometric properties in both its original [66] and French validated version [68]. It determines whether the voices are omnipotent, benevolent or malevolent. It also explores the subject's beliefs about the voices and their reaction to them, namely the acceptance of their discourse and/or the ability to resist them. Only patients presenting with AVHs at baseline were able to complete this questionnaire.

**Assessment of the social cognition marker.** Social cognition is assessed using the "Theory of Mind" and "Emotion Recognition" tests of the NEPSY II test [69], in French version [70]. The "Theory of Mind" test assesses the child's ability to understand other perspectives, intentions and beliefs. "Emotion Recognition" assesses the ability to recognize facial effects based on six facial expressions: joy, sadness, anger, fear, disgust and a neutral expression. The NEPSY-II Theory of Mind and Affect Recognition subtests have shown acceptable validity and reliability in pediatric populations [71,72].

All clinical interviews and psychometric assessments, including the K-SADS-PL, MINI-Kid, WISC-IV, NEPSY-II, BAVQ-R, and DES-IV, were administered by the same evaluator, a child and adolescent psychiatrist and PhD student, ensuring both clinical expertise and consistency across all assessments.

## Procedures

This study includes two stages: the screening and study assessment performed immediately after inclusion (T0), and a follow-up assessment performed after six months (T1). The study procedure are illustrated in Fig 1.

The study model does not allow blinding because it is a case-control study and the patient's clinical data contains the question of the presence or absence of AVH at each stage of the protocol. At inclusion, "case" patients were recruited first, and then "control" patients were selected to match them for age and sex. During this study, participants continue their usual child psychiatric follow-up, and other types of treatment can be offered.

**Recruitment and consent.** Recruitment was carried out by child psychiatrists from the University Department of Child and Adolescent Psychiatry of the University Pediatric Hospital-Lenval in Nice. Patients are referred to the study evaluator by the referring child psychiatrist. All participants are contacted individually by the experienced and trained evaluator. During screening, patients are informed of study procedures. Before participating in the evaluations, patients and their legal representatives provide signed informed consent.

**Inclusion visit (T0).** The inclusion visit was used to determine the inclusion and exclusion criteria for patients referred to the study. The K-SADS-PL and the MINI-Kid were administered, along with the abbreviated version of the WISC-IV. The AVH screening questionnaire was then completed to determine the group in which the patient would be assigned. Finally, emotional cognition tests (DES-IV and BAVQ-R) and social cognition tests (NEPSY-II) were administered.

**6-month follow-up visit (T1).** At the 6-month follow-up visit, the AVH screening questionnaire was administered again to all patients in the study to assess the evolution of AVHs, as well as the K-SADS-PL to evaluate changes in PTSD and psychotic disorder diagnoses, and the MINI-Kid.

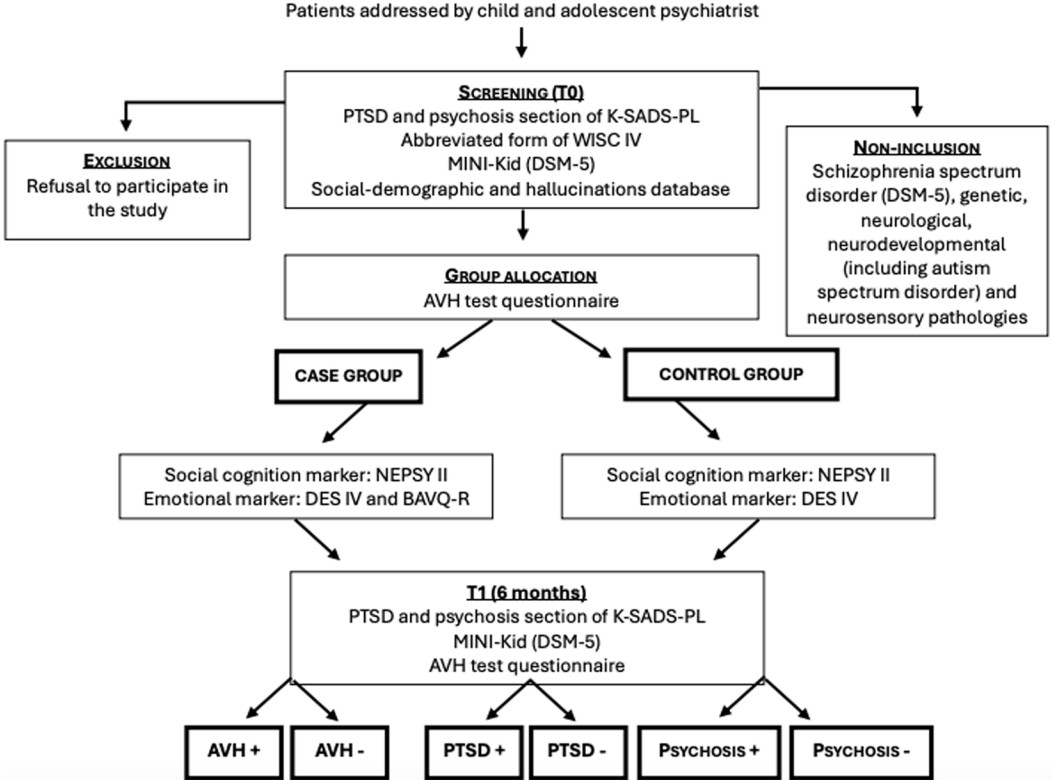

**Fig 1. Study procedure.**

We redistributed the comparison groups with and without AVHs between T0 and T1. At T0, the "case" and "control" groups corresponded to patients with and without AVHs at the time of inclusion in the study. At T1, the "HAV+" and "HAV–" groups were defined as follows: the "HAV+" group included patients from the initial "case" group whose AVHs persisted at 6 months or patients from the initial "control" group who developed AVHs at 6 months; the "HAV–" group included patients without AVHs at 6 months.

At T1, the patient groups are divided up again for comparison:

- The group of patients with AVH (persistent or appeared within 6 months) ("AVH +" group) and the group of patients without AVH (disappeared or never appeared within 6 months) ("AVH -" group).

- The group of patients with persistent PTSD at 6 months ("PTSD +" group) and the group of patients without PTSD at 6 months ("PTSD -" group).

- The group of patients with a psychotic disorder at 6 months ("Psychosis +" group) and the group of patients without a psychotic disorder at 6 months ("Psychosis -" group).

## Statistical analysis

The main objective of the study is a group comparison analysis looking for a significant difference between the "case" and "control" groups at T0. Continuous variables are compared using Student's t test (or the non-parametric Wilcoxon test if the variables do not follow a normal distribution) and concern the emotional cognition markers "DES IV" and "BAVQ-R".

Categorical variables are compared using the Chi$^2$ test (or Fisher's non-parametric two-tailed exact test if variables do not follow a normal distribution) and concern the NEPSY II social cognition markers "Theory of Mind" and "Emotion Recognition". Since there are two primary outcomes, correcting the alpha risk by the Bonferroni method is necessary, with an adjusted alpha of 2.5% for each marker to control the overall familywise error rate.

Concerning secondary objectives:

1) A new group comparison analysis is performed at T1 to compare markers of emotional and social cognition between groups with AVH at 6 months ("AVH +" group) and without AVH at 6 months ("AVH -" group).

2) A correlation analysis is performed between markers of emotional (DES IV and BAVQ-R) and social (NEPSY II) cognition and:

- the "case" and "control" groups at T0,

- then the "AVH +" and "AVH -" groups at T1.

Pearson's correlation method was used to measure the association between two continuous variables in a small population sample. For the correlation of categorical variables, the test of nullity of a correlation coefficient (or Spearman's method in the case where the variables do not follow a normal distribution) is used. The correlations between social cognition and emotional markers and the persistence of AVH in non-psychotic children and adolescents at 6 months are reassessed.

3) Emotional and social cognition markers are compared and correlated between:

- groups with PTSD ("PTSD+" group) and without PTSD ("PTSD-" group) at T1,

- then between groups with psychotic disorder ("Psychosis +" group) and without psychotic disorder ("Psychosis-" group) at T1.

### Ethical approval and consent

This study was approved by the Ethics Committee "Nord-Ouest III" (France) under protocol number 17-HPNCL-03. Written informed consent was obtained from all participants and/or their legal guardians prior to inclusion in the study.

## Results

### Study population

A total of 31 patients were included in the study: 16 patients in the "case" group and 15 patients in the "control" group at T0. The patient population included 80.6% girls and 19.4% boys, with a mean total age of 12.9 years (SD = 2.47). Fig 2 presents the types of trauma experienced by the participants, based on their self-report during the clinical interview (Fig 2). Fig 3 shows the distribution of groups at 6 months (T1) according to the evolution of AVH and the diagnoses of PTSD and psychotic disorder (DSM-5) (Fig 3).

### Assessment of markers of emotional and social cognition in AVH at T0 and T1 (Table 1)

**Analysis of results at T0.** At T0, social cognition scores measured with the NEPSY-II (Theory of Mind and Emotion Recognition subtests) and emotional cognition scores measured with the DES IV did not differ significantly between the "case" and "control" groups.

**Analysis of results at T1.** At T1, there was an evolution of AVH in both groups. In the "case" group, 9 patients still had AVH and 7 reported that they no longer had any. In the "control" group, 4 patients developed AVH within 6 months. A total

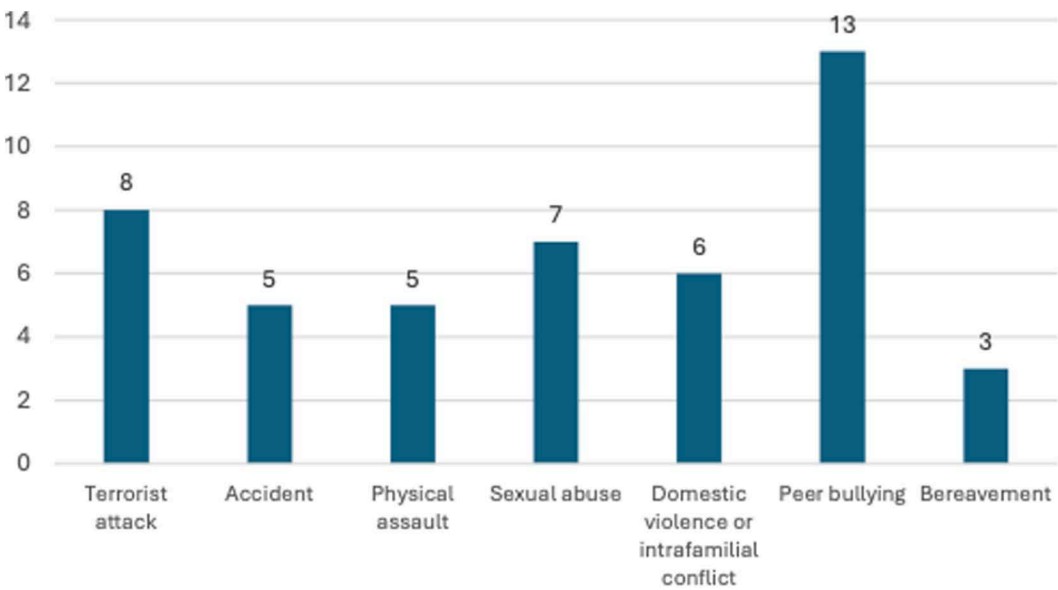

**Fig 2. Types of trauma experienced by the children included in the study.**

of 13 patients reported AVH and were combined in the "AVH +" group; 18 patients reported no AVH and were combined in the "AVH -" group (Fig 3). Statistical analyses revealed a significant difference and correlation for the emotional cognition marker. Participants with AVH at T1 had at T0, significantly higher malevolence scores in voices (BAVQ-R) and guilt levels (DES IV) compared with those without AVH at T1 (malevolence: p=0.04; guilt: p<0.05). Both malevolence and guilt scores at T0 were also significantly correlated with the presence of AVH at T1 (malevolence: r=0.36, p=0.04; guilt: r=0.35, p<0.05).

There were no significant results for markers of social cognition. The diagnosis of psychotic disorder at T1 was significantly related (p=0.01) and correlated (r=0.45, p=0.01) to the presence of AVH.

### Assessment of emotional and social cognition markers in PTSD at T0 and T1 (Table 2)

At T1, 19 patients still met the PTSD diagnosis and were combined in the "PTSD+" group; 12 patients no longer met the PTSD diagnosis and were combined in the "PTSD-" group (Fig 3). Statistical analyses showed that emotional cognition markers were significantly associated and correlated with PTSD progression at 6 months. Patients without a PTSD diagnosis at 6 months showed significantly omnipotent (p=0.01 and r=−0.46; p<0.01) and benevolent (p=0.01 and r=−0.44; p=0.01), as well as significant interactions with their AVHs, whether to resist them (p<0.05 and r=−0.36; p<0.04) or to adhere to their discourse emotionally (p=0.03 and r=−0.40; p=0.02) and behaviorally (p=0.01 and r=−0.45; p=0.01). Patients in the "PTSD+" group at T1 showed significant results for the emotions of disgust (p<0.01), fear (p<0.01) and anger (p<0.01). There were no significant differences or correlations for social cognition markers on PTSD progression at 6 months.

### Evaluation of markers of emotional and social cognition in psychotic disorder at T0 and T1 (Table 3)

At T1, 4 patients with a diagnosis of psychotic disorder (DSM-5) were grouped together in the "Psychosis +" group (Fig 3): all had significant AVH (p=0.02 and r=0.45; p=0.01) and a diagnosis of PTSD.

Statistical analyses of the "Psychosis +" and "Psychosis -" groups showed no significant relationship or correlation with markers of emotional and social cognition.

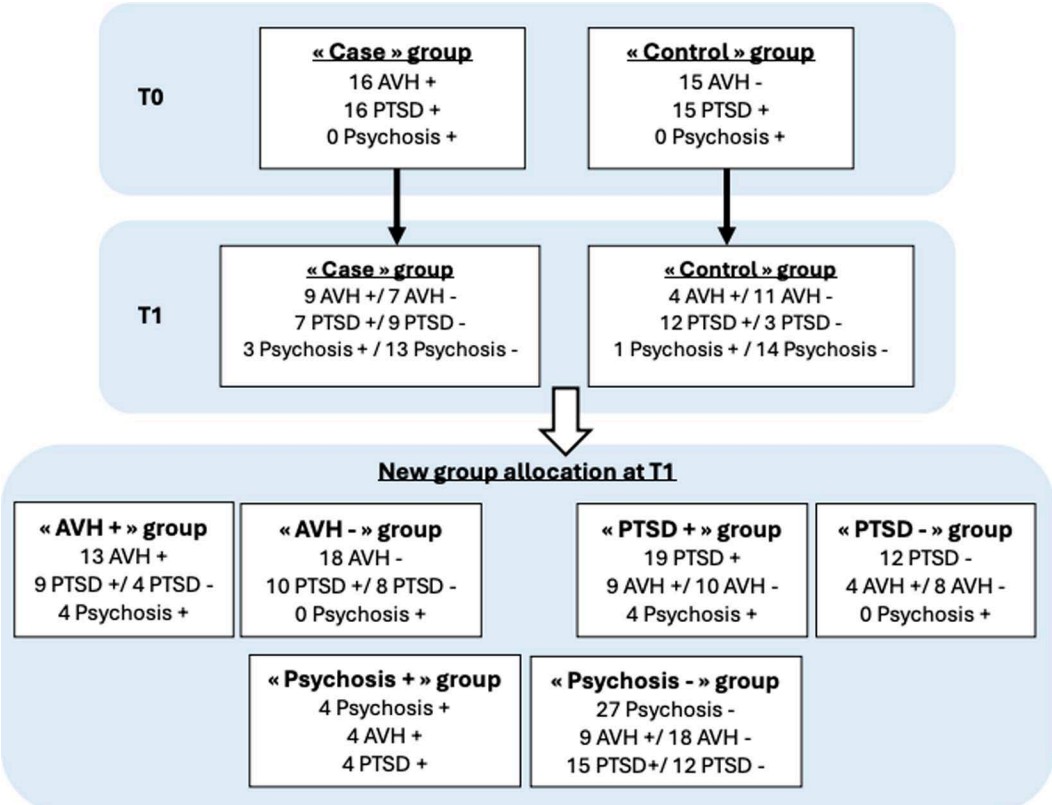

**Fig 3. Flow chart.**

## Discussion

Our study of children and adolescents with PTSD and AVH sought to identify markers of emotional and social cognition that would help us understand the presence of AVH and the development of post-traumatic and psychotic symptoms.

### Emotional impact on the development of hallucinations, PTSD and psychotic disorder

In our study, markers of emotional and social cognition were not associated with AVH present in patients with PTSD at inclusion. However, our results show that emotional cognitions are significantly associated with the presence of AVHs and the diagnosis of PTSD at their T1 reassessment. The malevolent of AVHs, reported by patients at T0, is significantly associated with their presence at T1. These results have already been found in our team's previous work [73,74]. The scientific literature also reports a negative content of hallucinations in a context of childhood adversity [75,76]. The negative emotional content of hallucinations (threatening, malevolent, hostile voices...) [77,78] is significantly found in patients with an associated psychiatric disorder [79–82] and in particular a psychotic disorder [83]. Guilt is also found in our results to be significantly associated with the presence of AVH at T1 and concurs with other work describing, after trauma, the emotions of shame and guilt that could both encourage the development of AVH and influence their content [84–86].

Our results find that emotions such as disgust, fear and anger are significantly associated with the persistence of the PTSD diagnosis at T1. These emotions are particularly found in the emotional dysregulation present in PTSD [40,87,88] and associated with the risk of progression to comorbid psychiatric pathologies [89,90].

**Table 1. Statistical analysis of "case" and "control" groups at T0 and "AVH +" and "AVH –" groups at T1.**

| | T0 | | | | | | T1 | | | | | |
|---|---|---|---|---|---|---|---|---|---|---|---|---|
| | Total (N=31) | « case » group (N=16) | « control » group (N=15) | p | Correlation coefficient (r) | p | Total (N=31) | « AVH+ » group (N=13) | « AVH - » group (N=18) | p | Correlation coefficient (r) | p |
| Sex (%) | | | | 0,92 | 0,01 | 0,93 | | | | 0,63 | 0,08 | 0,65 |
| • Male | 6 (19,4) | 3 (18,8) | 3 (20) | | | | 6 (19,4) | 2 (15,4) | 4 (22,2) | | | |
| • Female | 25 (80,6) | 13 (81,2) | 12 (80) | | | | 25 (80,6) | 11 (84,6) | 14 (77,8) | | | |
| Age (mean) | 12,90 | 12,75 | 13,07 | 0,68 | −0,07 | 0,68 | 12,90 | 13,15 | 12,7 | 0,49 | 0,13 | 0,49 |
| PTSD + (%) | 31 (100) | 16 (100) | 15 (100) | 1 | 0 | 1 | 19 | 9 (69,2) | 10 | 0,44 | 0,14 | 0,48 |
| Psychosis + (%) | 0 | 0 | 0 | 0 | 0 | 0 | 4 (12,9) | 4 (30,8) | 0 | **0,01\*** | 0,45 | **0,01\*** |
| BAVQ-R (mean) | | | | | | | | | | | | |
| Omnipotence | | 18,69 | 0 | < 0,001 | 0,92 | < 0,001 | 9,64 | 12,92 | 7,28 | 0,15 | 0,24 | 0,15 |
| Benevolence | | 8,68 | 0 | < 0,001 | 0,92 | < 0,001 | 4,48 | 6,46 | 3,06 | 0,09 | 0,31 | 0,08 |
| Malevolence | | 16,81 | 0 | < 0,001 | 0,92 | < 0,001 | 8,68 | 12,23 | 6,11 | **0,04\*** | 0,36 | **0,04\*** |
| Emotional resistance | | 12,75 | 0 | < 0,001 | 0,92 | < 0,001 | 6,58 | 9,23 | 4,67 | 0,07 | 0,33 | 0,07 |
| Behavioral resistance | | 17,0 | 0 | < 0,001 | 0,92 | < 0,001 | 8,77 | 12,38 | 6,17 | 0,07 | 0,33 | 0,07 |
| Emotional engagement | | 5,0 | 0 | < 0,001 | 0,95 | < 0,001 | 6,58 | 3,46 | 1,94 | 0,08 | 0,32 | 0,07 |
| Behavioral resistance | | 5,25 | 0 | < 0,001 | 0,93 | < 0,001 | 8,77 | 3,92 | 1,83 | 0,10 | 0,30 | 0,09 |
| Theory of mind (%) | | | | 0,86 | −0,06 | 0,74 | | | | 0,20 | 0,06 | 0,73 |
| • Normal score | 15 (48,4) | 8 (50) | 7 (46,6) | | | | 15 (48,4) | 5 (38,5) | 10 (55,6) | | | |
| • Lower normal score | 9 (29) | 5 (31,25) | 4 (26,6) | | | | 9 (29) | 6 (46,1) | 3 (16,7) | | | |
| • Lower anormal score | 7 (22,6) | 3 (18,75) | 4 (26,6) | | | | 7 (22,6) | 2 (15,4) | 5 (27,8) | | | |
| Emotion Recognition (%) | | | | 0,32 | 0,18 | 0,34 | | | | 0,23 | 0,21 | 0,24 |
| • Normal score | 30 (96,2) | 15 (93,75) | 15 (100) | | | | 30 (96,2) | 12 (92,3) | 18 (100) | | | |
| • Lower normal score | 1 (3,8) | 1 (6,25) | 0 | | | | 1 (3,8) | 1 (7,7) | 0 | | | |
| • Lower anormal score | 0 | 0 | 0 | | | | 0 | 0 | 0 | | | |
| Recognition of joy | | | | 0,84 | 0 | 1 | | | | 0,60 | 0,11 | 0,54 |
| • Normal score | 17 (54,8) | 9 (56,25) | 8 (53,3) | | | | 17 (54,8) | 6 (46,1) | 11 5 (61,1) | | | |
| • Lower normal score | 9 (29) | 4 (25) | 5 (33,3) | | | | 9 (29) | 5 (38,5) | 4 (22,2) | | | |
| • Lower anormal score | 5 (16,2) | 3 (18,75) | 2 (13,3) | | | | 5 (16,2) | 2 (15,4) | 3 (16,7) | | | |
| Recognition of sadness | | | | 0,66 | −0,16 | 0,38 | | | | 0,35 | −0,26 | 0,16 |
| • Normal score | 17 (54,8) | 10 (62,5) | 7 (46,6) | | | | 17 (54,8) | 9 (63,2) | 8 (44,4) | | | |
| • Lower normal score | 9 (29) | 4 (25) | 5 (33,4) | | | | 9 (29) | 3 (23,1) | 6 (33,3) | | | |
| • Lower anormal score | 5 (16,2) | 2 (12,5) | 3 (20) | | | | 5 (16,2) | 1 (7,7) | 4 (22,2) | | | |
| Recognition of neutral emotion | | | | 0,66 | 0,01 | 0,94 | | | | 0,65 | 0,08 | 0,66 |
| • Normal score | 20 (64,5) | 10 (62,5) | 10 (66,6) | | | | 20 (64,5) | 8 (61,5) | 12 (66,6) | | | |
| • Lower normal score | 8 (25,8) | 5 (31,25) | 3 (20) | | | | 8 (25,8) | 3 (23,1) | 5 (27,7) | | | |
| • Lower anormal score | 3 (9,7) | 1 (6,25) | 2 (13,3) | | | | 3 (9,7) | 2 (15,4) | 1 (5,5) | | | |

*(Continued)*

**Table 1.** (Continued)

| | T0 | | | | | | T1 | | | | | |
|---|---|---|---|---|---|---|---|---|---|---|---|---|
| | Total (N=31) | «case» group (N=16) | «control» group (N=15) | p | Correlation coefficient (r) | p | Total (N=31) | «AVH+» group (N=13) | «AVH-» group (N=18) | p | Correlation coefficient (r) | p |
| Recognition of fear | | | | 0,60 | 0,14 | 0,43 | | | | 0,86 | 0,09 | 0,61 |
| • Normal score | 23 (74,2) | 11 (68,75) | 12 (80) | | | | 23 (74,2) | 9 (69,2) | 14 (77,8) | | | |
| • Lower normal score | 4 (12,9) | 2 (12,5) | 2 (13,3) | | | | 4 (12,9) | 2 (15,4) | 2 (11,1) | | | |
| • Lower anormal score | 4 (12,9) | 3 (18,75) | 1 (6,7) | | | | 4 (12,9) | 2 (15,4) | 2 (11,1) | | | |
| Recognition of anger | | | | 0,56 | −0,04 | 0,84 | | | | 0,28 | 0,26 | 0,16 |
| • Normal score | 23 (74,2) | 12 (75) | 11 (73,3) | | | | 23 (74,2) | 8 (61,5) | 15 (83,3) | | | |
| • Lower normal score | 7 (22,6) | 4 (25) | 3 (20) | | | | 7 (22,6) | 4 (30,8) | 3 (16,7) | | | |
| • Lower anormal score | 1 (3,2) | 0 | 1 (6,7) | | | | 1 (3,2) | 1 (7,7) | 0 | | | |
| Recognition of disgust | | | | 0,37 | −0,22 | 0,24 | | | | 0,71 | −0,09 | 0,60 |
| • Normal score | 22 (70,9) | 13 (81,25) | 9 (60) | | | | 22 (70,9) | 10 (76,9) | 12 (66,7) | | | |
| • Lower normal score | 7 (22,6) | 2 (12,5) | 5 (33,3) | | | | 7 (22,6) | 2 (15,4) | 5 (27,7) | | | |
| • Lower anormal score | 2 (6,4) | 1 (6,25) | 1 (6,7) | | | | 2 (6,4) | 1 (7,7) | 1 (5,5) | | | |
| DES IV (mean) | | | | | | | | | | | | |
| Guilt | 8,97 | 9,88 | 8,00 | 0,09 | 0,31 | 0,08 | 8,97 | 10,31 | 8,00 | <0,05* | 0,35 | <0,05* |
| Shyness | 10,35 | 10,75 | 9,83 | 0,55 | 0,11 | 0,54 | 10,35 | 9,54 | 10,94 | 0,28 | −0,20 | 0,28 |
| Joy | 8,06 | 8,38 | 7,73 | 0,63 | 0,09 | 0,63 | 8,06 | 8,31 | 7,89 | 0,63 | 0,09 | 0,62 |
| Disgust | 8,45 | 8,75 | 8,13 | 0,75 | 0,06 | 0,74 | 8,45 | 8,23 | 8,61 | 0,73 | −0,06 | 0,72 |
| Hostility against self | 10,16 | 10,94 | 9,33 | 0,16 | 0,26 | 0,16 | 10,16 | 10,38 | 10,00 | 0,85 | 0,04 | 0,84 |
| Shame | 11,71 | 12,12 | 11,27 | 0,53 | 0,11 | 0,53 | 11,71 | 11,54 | 11,83 | 0,70 | −0,07 | 0,69 |
| Sadness | 11,65 | 12,06 | 11,20 | 0,54 | 0,11 | 0,54 | 11,65 | 11,85 | 11,50 | 1 | −0,003 | 0,98 |
| Surprise | 8,39 | 9,00 | 7,73 | 0,22 | 0,23 | 0,22 | 8,39 | 8,69 | 8,17 | 0,59 | 0,09 | 0,59 |
| Contempt | 5,68 | 5,69 | 5,67 | 0,94 | 0,02 | 0,92 | 5,68 | 5,46 | 5,83 | 0,50 | −0,13 | 0,50 |
| Interest | 8,52 | 8,31 | 8,73 | 0,72 | −0,06 | 0,71 | 8,52 | 8,92 | 8,22 | 0,48 | 0,13 | 0,47 |
| Fear | 10,42 | 10,81 | 10,00 | 0,84 | 0,04 | 0,83 | 10,42 | 10,31 | 10,50 | 0,92 | −0,02 | 0,90 |
| Anger | 10,39 | 10,44 | 10,33 | 0,96 | 0,01 | 0,95 | 10,39 | 9,85 | 10,78 | 0,52 | −0,12 | 0,51 |

**Table 2. Statistical analysis of "PTSD +" and "PTSD -" groups at T1.**

| | T1 | | | | |
| --- | --- | --- | --- | --- | --- |
| | PTSD +<br>(N = 19) | PTSD –<br>(N = 12) | p | Correlation coefficient (r) | p |
| Sex (%) | | | | | |
| • Male | 3 (15,8) | 3 (25) | 0,53 | 0,11 | 0,54 |
| • Female | 16 (84,2) | 9 (75) | | | |
| Age (mean) | 13,05 | 12,67 | 0,98 | 0,007 | 0,96 |
| AVH + at T1 (%) | 9 (47,4) | 4 (33,3) | 0,44 | 0,14 | 0,45 |
| Psychosis at T1 (%) | 4 (21,5) | 0 | 0,09 | 0,30 | 0,09 |
| BAVQ-R (mean) | | | | | |
| Omnipotence | 6,58 | 14,50 | **0,01*** | −0,46 | **< 0,01*** |
| Benevolence | 2,37 | 7,83 | **0,01*** | −0,44 | **0,01*** |
| Malevolence | 7,16 | 11,08 | 0,24 | −0,21 | 0,24 |
| Emotional resistance | 5,26 | 8,67 | 0,20 | −0,23 | 0,19 |
| Behavioral resistance | 6,05 | 13,08 | **<0,05*** | −0,36 | **0,04*** |
| Emotional engagement | 1,68 | 4 | **0,03*** | −0,40 | **0,02*** |
| Behavioral engagement | 1,89 | 4 | **0,01*** | −0,45 | **0,01*** |
| Theory of mind (%) | | | 0,36 | −0,06 | 0,73 |
| • Normal score | 9 (47,4) | 6 (50) | | | |
| • Lower normal score | 7 (36,8) | 2 (16,7) | | | |
| • Lower anormal score | 3 (15,8) | 4 (33,3) | | | |
| Emotion Recognition (%) | | | 0,20 | −0,23 | 0,21 |
| • Normal score | 19 (100) | 11 (91,7) | | | |
| • Lower normal score | 0 | 1 (8,3) | | | |
| • Lower anormal score | 0 | 0 | | | |
| Recognition of joy | | | 0,39 | −0,008 | 0,96 |
| • Normal score | 11 (57,9) | 6 (50) | | | |
| • Lower normal score | 4 (21,05) | 5 (41,7) | | | |
| • Lower anormal score | 4 (21,05) | 1 (8,3) | | | |
| Recognition of sadness | | | 0,92 | 0,04 | 0,82 |
| • Normal score | 10 (52,6) | 7 (58,3) | | | |
| • Lower normal score | 6 (31,6) | 3 (25) | | | |
| • Lower anormal score | 3 (15,8) | 2 (16,7) | | | |
| Recognition of neutral emotion | | | 0,60 | 0,16 | 0,38 |
| • Normal score | 11 (57,9) | 9 (75) | | | |
| • Lower normal score | 6 (31,6) | 2 (16,7) | | | |
| • Lower anormal score | 2 (10,5) | 1 (8,3) | | | |
| Recognition of fear | | | 0,76 | −0,004 | 0,97 |
| • Normal score | 14 (73,7) | 9 (75) | | | |
| • Lower normal score | 3 (15,8) | 1 (8,3) | | | |
| • Lower anormal score | 2 (10,5) | 2 (16,7) | | | |
| Recognition of anger | | | 0,56 | 0,17 | 0,34 |
| • Normal score | 13 (68,4) | 10 (83,3) | | | |
| • Lower normal score | 5 (26,3) | 2 (16,7) | | | |
| • Lower anormal score | 1 (5,2) | 0 | | | |
| Recognition of disgust | | | 0,99 | 0,05 | 0,76 |
| • Normal score | 13 (68,4) | 9 (75) | | | |

*(Continued)*

**Table 2.**  (Continued)

| | T1 | | | | |
|---|---|---|---|---|---|
| | PTSD +<br>(N = 19) | PTSD −<br>(N = 12) | p | Correlation coefficient (r) | p |
| • Lower normal score | 5 (26,3) | 2 (16,7) | | | |
| • Lower anormal score | 1 (5,2) | 1 (8,3) | | | |
| DES IV (mean) | | | | | |
| Guilt | 9,21 | 8,58 | 0,46 | 0,14 | 0,46 |
| Shyness | 11,16 | 9,08 | 0,13 | 0,28 | 0,13 |
| Joy | 7,32 | 9,25 | 0,20 | −0,23 | 0,20 |
| Disgust | 9,79 | 6,33 | **< 0,01*** | 0,51 | **< 0,01*** |
| Hostility against self | 11,16 | 8,58 | 0,12 | 0,29 | 0,12 |
| Shame | 12,47 | 10,50 | 0,07 | 0,33 | 0,07 |
| Sadness | 12,89 | 9,67 | 0,09 | 0, 31 | 0,09 |
| Surprise | 8,68 | 7,92 | 0,25 | 0,21 | 0,25 |
| Contempt | 5,79 | 5,50 | 0,85 | 0,03 | 0,84 |
| Interest | 9,26 | 7,33 | 0,09 | 0,30 | 0,09 |
| Fear | 12,11 | 7,75 | **< 0,01*** | 0,58 | **< 0,001*** |
| Anger | 11,89 | 8,00 | **< 0,01*** | 0,55 | **< 0,001*** |

## Impact of PTSD on the emotional valence of AVHs

The results of our study show that the presence of AVH at T1 is significantly associated with the perception of malicious hallucinations on the BAVQ-R scale. We also found that patients who no longer had PTSD at T1 significantly perceived these hallucinations as benevolent. These findings are in line with the literature describing the negative content of hallucinations in a trauma context [75,76]. This is described in particular by the fact that hallucinations and trauma share the same negative theme [86,91–94]. Hallucinations here can be seen as the return of fragments of intrusive memories that are non-contextualized and attributed to a source external to the subject [95].

Although AVHs are perceived as benevolent in the subject without PTSD at T1, they nevertheless remain described by the patients in the study as significantly omnipotent in the bond they maintain with it, whether to fight against them or adhere to their discourses. This is in line with the concept described in the literature that AVHs are a defense mechanism for externalizing internal conflicts that are difficult for the subject to mentalize, particularly in children and adolescents with psychic immaturity [10,13,96,97]. AVHs would represent an avoidance strategy for intrusive symptoms by expressing traumatic memories in a different sensory modality [98]. Hallucinations would thus protect patients from PTSD symptoms by encouraging self-protective behavior against them [99].

## Evolution of post-traumatic symptoms

Finally, in this study, we found that a 6-month reassessment enabled us to observe an evolution in the symptoms of the patients included: of the 16 patients with AVH at T0, 13 had them at T1, including 4 patients who did not have them at the time of their inclusion in the study. This highlights that hallucinatory experience in the context of psycho-trauma is an evolving symptom [10,50]. Our results highlight a significant link between AVH in PTSD and the development of a psychotic disorder. This is in line with our initial hypothesis placing hallucinatory experience as an evolutionary step in the continuum between psychic trauma and psychotic disorder [87,100]. The literature describes this psychopathological model on the basis of mainly retrospective studies [101]. These findings may suggest a psychopathological framework: in children and adolescents, the presence of AVH could warrant systematic exploration of trauma history, as traumatic

**Table 3. Statistical analyses of "Psychosis +" and "Psychosis -" groups at T0 and T1.**

| | T1 | | | | |
| --- | --- | --- | --- | --- | --- |
| | Psychosis+ (N = 4) | Psychosis – (N = 27) | p | Correlation coefficient (r) | p |
| Sex (%) | | | 0,09 | −0,30 | 0,10 |
| • Male | 2 (50) | 4 (14,8) | | | |
| • Female | 2 (50) | 23 (85,2) | | | |
| Age (mean) | 13 | 12,89 | 0,97 | −0,01 | 0,95 |
| AVH + at T1 (%) | 4 (100) | 9 (33,3) | **0,02*** | 0,45 | **0,01*** |
| PTSD + at T1 (%) | 4 (100) | 15 (55,6) | 0,09 | 0,30 | 0,09 |
| BAVQ-R (mean) | | | | | |
| Omnipotence | 13 | 9,15 | 0,68 | 0,08 | 0,66 |
| Benevolence | 4,75 | 4,44 | 0,63 | 0,09 | 0,62 |
| Malevolence | 14,50 | 7,81 | 0,21 | 0,23 | 0,20 |
| Emotional resistance | 10,75 | 5,96 | 0,21 | 0,23 | 0,20 |
| Behavioral resistance | 13,75 | 8,04 | 0,24 | 0,22 | 0,24 |
| Emotional engagement | 3 | 2,52 | 0,60 | 0,10 | 0,60 |
| Behavioral engagement | 3 | 2,67 | 0,82 | 0,05 | 0,80 |
| Theory of mind (%) | | | 0,08 | 0,04 | 0,80 |
| • Normal score | 1 (25) | 14 (51,8) | | | |
| • Lower normal score | 3 (75) | 6 (22,2) | | | |
| • Lower anormal score | 0 | 7 (25,9) | | | |
| Emotion Recognition (%) | | | 0,70 | −0,07 | 0,70 |
| • Normal score | 4 (100) | 26 (96,3) | | | |
| • Lower normal score | 0 | 1 (3,7) | | | |
| • Lower anormal score | 0 | 0 | | | |
| Recognition of joy | | | 0,39 | −0,09 | 0,60 |
| • Normal score | 3 (75) | 14 (51,8) | | | |
| • Lower normal score | 0 | 9 (33,3) | | | |
| • Lower anormal score | 1 (25) | 4 (14,8) | | | |
| Recognition of sadness | | | 0,15 | −0,33 | 0,06 |
| • Normal score | 4 (100) | 13 (48,1) | | | |
| • Lower normal score | 0 | 9 (33,3) | | | |
| • Lower anormal score | 0 | 5 (18,5) | | | |
| Recognition of neutral emotion | | | 0,77 | −0,10 | 0,57 |
| • Normal score | 3 (75) | 17 (63) | | | |
| • Lower normal score | 1 (25) | 7 (25,9) | | | |
| • Lower anormal score | 0 | 3 (11,1) | | | |
| Recognition of fear | | | 0,45 | −0,22 | 0,22 |
| • Normal score | 4 (100) | 19 (70,4) | | | |
| • Lower normal score | 0 | 4 (14,8) | | | |
| • Lower anormal score | 0 | 4 (14,8) | | | |
| Recognition of anger | | | 0,36 | 0,20 | 0,29 |
| • Normal score | 2 (50) | 21 (77,7) | | | |
| • Lower normal score | 2 (50) | 5 (18,5) | | | |
| • Lower anormal score | 0 | 1 (3,7) | | | |

*(Continued)*

**Table 3.** (Continued)

| | T1 | | | | |
|---|---|---|---|---|---|
| | Psychosis+ (N = 4) | Psychosis − (N = 27) | p | Correlation coefficient (r) | p |
| Recognition of disgust | | | 0,39 | −0,24 | 0,19 |
| • Normal score | 4 (100) | 18 (66,7) | | | |
| • Lower normal score | 0 | 7 (25,9) | | | |
| • Lower anormal score | 0 | 2 (7,4) | | | |
| DES IV (mean) | | | | | |
| Guilt | 10 | 8,81 | 0,40 | 0,16 | 0,40 |
| Shyness | 7 | 10,85 | 0,06 | −0,34 | 0,06 |
| Joy | 8,75 | 7,96 | 0,55 | 0,11 | 0,54 |
| Disgust | 8,50 | 8,44 | 0,97 | −0,01 | 0,95 |
| Hostility against self | 11,25 | 10,00 | 0,61 | 0,10 | 0,60 |
| Shame | 11,50 | 11,74 | 0,61 | −0,10 | 0,60 |
| Sadness | 11,50 | 11,67 | 0,97 | −0,01 | 0,95 |
| Surprise | 8,25 | 8,41 | 0,86 | −0,03 | 0,8' |
| Contempt | 6,75 | 5,52 | 0,45 | 0,15 | 0,43 |
| Interest | 10,25 | 8,26 | 0,44 | 0,23 | 0,22 |
| Fear | 11,50 | 10,26 | 0,48 | 0,13 | 0,47 |
| Anger | 11,25 | 10,26 | 0,61 | 0,10 | 0,60 |

experiences may contribute to the content of AVH. Conversely, when PTSD is diagnosed, close monitoring may help identify the possible emergence of perceptual disturbances such as hallucinations, or even progression toward a psychotic disorder. This proposed framework is in line with literature suggesting that psychiatric symptoms often evolve dynamically over time [102,103], gradually leading to comorbid conditions [22,104] and supporting a network-based understanding of psychopathology [105,106]. The originality of this study lies in the prospective analysis of symptom progression, highlighting markers of emotional cognition as a risk factor for poor progression and the target of specific preventive care.

## Limitations

This study is a follow-up to an initial pilot study on AVH in non-psychotic children and adolescents. This study was conducted at a single center on a small patient sample (N = 31), which limits the robustness of the conclusions when comparing groups with AVHs (N = 16) and without AVHs (N = 15) over a short follow-up period (6 months), thereby complicating the interpretation of the results. Moreover, this study includes patients covering a wide developmental age range, during which hallucinatory symptomatology shows clinical and prognostic variability. The standard deviation of 2.47 for a mean age of 12.9 years remains moderate in this context. The theme of these studies, although particularly important for understanding the psychopathological mechanisms at play between psycho-trauma, hallucinatory experience and psychotic disorder, presents feasibility difficulties. Hallucinations in non-psychotic children and adolescents are a symptomatology not easily accessible to untrained clinicians. Screening must be targeted, with specific clinical features on questioning to clearly differentiate traumatic reliving from hallucinations. Also, the number of patients included in the study remains small sample sizes.

This study did not include a comparison between the AVH+ and AVH− groups regarding the objective nature or severity of the traumatic events, as our inclusion criterion was based on meeting DSM-5 diagnostic criteria for PTSD, which emphasizes the subjective experience of trauma [107]. In addition, no data were collected on the exact time elapsed

between the traumatic event and study inclusion, particularly as many participants reported complex or repeated trauma exposures over time, and PTSD decompensation could occur following a later event rather than the initial one. These factors should be considered as limitations of the present study.

In our study, no significant differences were observed for social cognition markers. Previous studies, however, have reported impairments in social cognition among youth at risk of psychosis and in PTSD populations, suggesting a potential mediating role of social cognition between trauma exposure and subsequent psychopathology, even before the onset of the first psychotic symptoms [46,47,108,109]. At the same time, other studies have not retained social cognition as a consistent vulnerability marker for psychotic symptomatology [110], supporting instead the hypothesis of neurodevelopmental differences in social cognition between individuals progressing to schizophrenia spectrum disorders and those experiencing only isolated psychotic symptoms [111,112]. Furthermore, we can discuss the choice of assessment tools for emotional and social cognition markers in our studies. There are still few validated tools for assessing emotional and social cognition in the pediatric population [113]. The DES-IV and the BAVQ-R are not validated in pediatric populations, and the NEPSY-II presents certain limitations, particularly in the way tasks are administered [103]. All assessments were conducted by an expert examiner, allowing adaptation to the characteristics of the study population. Finally, scientific work proposing a developmental approach to the impairment of social cognition in psycho-trauma reports that this impairment would depend on the conditions of the event, the moment when it took place contemporary with a "pivotal" period in the child's cognitive development, and the length of time needed to objectify cognitive impairment after the trauma [108,114]. It will thus be necessary to complete the study to determine the association of social cognition on AVH in children and adolescents with PTSD.

## Conclusion

Markers of emotional cognition are significantly associated with the presence and persistence of AVH in children and adolescents with PTSD, as well as in the course of post-traumatic and psychotic symptoms. Emotional cognition acts as a common marker for PTSD and hallucinatory experience, reinforcing the notion of a continuum between psycho-trauma and psychotic disorder. This prospective study highlights both fluctuating post-traumatic and hallucinatory symptomatology, and a highly significant association between AVH and the risk of progressing to a diagnosis of psychosis at 6 months. Thus, while post-traumatic symptomatology may not be initially fixed, the presence of AVH may represent a risk factor for poor evolution towards a psychotic disorder. Emotional cognition is thus emerging as a target area for psychopathological and therapeutic understanding in patients with PTSD, to prevent progression to psychotic symptoms.

## Supporting information

**S1 File.  BBD Physalis 2.**
(XLSX)

## Acknowledgments

The authors would like to acknowledge the contributors of the study, Foundation Lenval, Pediatric Hospitals of Nice CHU-Lenval and CoBTeK Laboratory and the Nice Pediatric Psychotrauma Center (NPPC).

## Author contributions

**Conceptualization:** Louise-Emilie Dumas.

**Data curation:** Louise-Emilie Dumas.

**Formal analysis:** Louise-Emilie Dumas.

**Investigation:** Louise-Emilie Dumas.

**Methodology:** Louise-Emilie Dumas, Florence Askenazy, Arnaud Fernandez.

**Resources:** Louise-Emilie Dumas.

**Software:** Louise-Emilie Dumas.

**Supervision:** Florence Askenazy, Arnaud Fernandez.

**Validation:** Louise-Emilie Dumas, Arnaud Fernandez.

**Visualization:** Louise-Emilie Dumas.

**Writing – original draft:** Louise-Emilie Dumas.

**Writing – review & editing:** Louise-Emilie Dumas, Florence Askenazy, Arnaud Fernandez.

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
