## [Decision Letter · Decision Letter 0]

6 Aug 2025

PONE-D-25-30151Relationships between markers of emotional and social cognition and acoustic-verbal hallucinations in children and adolescents with post-traumatic stress disorder (PTSD)PLOS ONE

Dear Dr. Fernandez,

Thank you for submitting your manuscript to PLOS ONE. After careful consideration, we feel that it has merit but does not fully meet PLOS ONE’s publication criteria as it currently stands. Therefore, we invite you to submit a revised version of the manuscript that addresses the points raised during the review process.

We look forward to receiving your revised manuscript.

Kind regards,

Mu-Hong Chen, M.D., Ph.D.

Academic Editor

PLOS ONE

Journal Requirements:

2. In the online submission form, you indicated that “Summary and anonymized data would be available on the basis of a written reasonable request”.

4. Please include your tables as part of your main manuscript and remove the individual files. Please note that supplementary tables (should remain/ be uploaded) as separate "supporting information" files.

Reviewers' comments:

Reviewer's Responses to Questions

**Comments to the Author**

1. Is the manuscript technically sound, and do the data support the conclusions?

Reviewer #1: Partly

Reviewer #2: Yes

2. Has the statistical analysis been performed appropriately and rigorously? 

Reviewer #1: I Don't Know

Reviewer #2: Yes

3. Have the authors made all data underlying the findings in their manuscript fully available?

Reviewer #1: Yes

Reviewer #2: Yes

4. Is the manuscript presented in an intelligible fashion and written in standard English?

Reviewer #1: Yes

Reviewer #2: Yes

5. Review Comments to the Author

Reviewer #1: Thank you for the opportunity to review this interesting manuscript, which explores the relationship between the presence of AVH (acoustic-verbal hallucinations) and emotional as well as social cognition in children and adolescents. As a clinician, this is a symptom we frequently encounter in practice, and I am also curious about its onset and association with the clinical course.

To begin with, although the results are promising, I noticed that the tables were not included in the PDF file. This made it somewhat difficult to accurately interpret the findings. If possible, I would appreciate the opportunity to review the full version with the tables included.

In addition, I would like to raise a few questions regarding definitions and methodological points:

1.I am curious as to why the term “acoustic-verbal hallucinations (AVH)” was chosen instead of directly referring to “auditory hallucinations with voice commanding.”

2.Regarding the statement that “no difference was found in emotional and social cognition at baseline,” does this refer to no between-group differences? Since the mean scale scores were not reported, I wonder whether both groups (being PTSD patients) showed similarly impaired emotional and social cognition at baseline; and perhaps only after six months did group differences emerge due to improvement in some individuals?

3.Given that peer bullying constituted a relatively high proportion of trauma types in the sample, I would appreciate clarification on how "peer bullying" was defined in this study.

4.In Figure 3, there appears to be a typographical error at T0: the label “control group” should be “AVH–.”

Reviewer #2: This study examines whether the emotional and social cognition of children and adolescents with PTSD can predict subsequent AVH, in what may be a follow-up study lasting up to seven years. This is an important issue in terms of early monitoring of the psychopathological course following trauma and may also help clarify the psychopathological underpinnings of AVH in PTSD. Suggestions and explanations are as follows.

1. Suggested revision to phrasing of study objectives:

The current wording—“The primary objective of our study” and “The secondary objectives of our study”—may imply that the secondary objectives are unrelated. However, the listed secondary objectives appear to elaborate on and further specify the primary objective. Therefore, I suggest revising “The secondary objectives of our study are” to “Specifically, we aim to” or “To this end, we further aim to”, which would better reflect their relationship.

2. Clarification on trauma exposure:

Although all participants met the DSM-5 criteria for PTSD, was there a comparison between the AVH and non-AVH groups in terms of the nature or severity of the traumatic events they experienced? Were the stressors similar across groups, or were there meaningful differences?

Additionally, did the two groups differ in the time interval between the occurrence of the stressful event and their inclusion in the study? The current mental state of children and adolescents may also be influenced by how much time has passed since the traumatic event.

3. Diagnostic validity and assessor qualifications:

Diagnostic accuracy is critical. Who administered the K-SADS-PL and the MINI-Kid (Mini International Neuropsychiatric Interview for Children and Adolescents)? Did these assessors receive formal training or hold any relevant professional licenses or certifications?

4. WISC-IV administration:

Who administered the WISC-IV? Did the examiners receive standardized training or hold professional certification to ensure the reliability and validity of the assessment?

5. Psychometric properties of measures:

It would strengthen the study to include more detailed information on the reliability and validity of each test used, particularly those assessing emotional and social cognition.

6. Literature support for null findings:

The statement “This study did not find significant results for markers of social cognition, contrary to data reported in the scientific literature” implies a contradiction with previous findings. Could the authors cite specific studies or provide references to support this comparison?

7. IQ differences between groups:

The study reports no significant differences in emotional and social cognition between AVH and non-AVH groups at T0. Given that IQ was measured, were there any differences in IQ between these two groups that might account for the findings?

8. Need for a non-PTSD comparison group:

The study hypothesizes that AVH emerging after PTSD may be due to impairments in emotional and social cognition, which affect individuals’ interpretations of internal neural signals. However, to support this hypothesis, it may be necessary to include a matched comparison group without PTSD. If both the AVH and non-AVH groups show lower emotional and social cognition scores than the control group—despite all three groups having similar IQs—it would provide a stronger foundation for attributing AVH to post-traumatic deficits in emotional and social cognition. If such a comparison is not currently available, it is recommended to address this as a limitation of the study.

6. PLOS authors have the option to publish the peer review history of their article (what does this mean? ). If published, this will include your full peer review and any attached files.

**Do you want your identity to be public for this peer review?** For information about this choice, including consent withdrawal, please see our Privacy Policy .

Reviewer #1: No

Reviewer #2: **Yes: ** H.-H. Lu

---

## [Author Response · Author response to Decision Letter 1]

19 Aug 2025

Dear Academic Editor and Reviewers,

We would like to thank you for the careful reading of our manuscript entitled “Relationships between markers of emotional and social cognition and acoustic-verbal hallucinations in children and adolescents with post-traumatic stress disorder (PTSD)” by Louise-Emilie Dumas, Florence Askenazy, and Arnaud Fernandez, and for the constructive comments and suggestions you have provided. We greatly appreciate the time and effort you have dedicated to helping us improve our work.

We have carefully considered each of your remarks and have revised the manuscript accordingly. In the following pages, we respond point-by-point to the comments raised by the Academic Editor and each reviewer. All changes in the revised version are highlighted using the “Track Changes” mode.

We believe that these revisions have strengthened the manuscript and clarified key aspects of our study, both in terms of methodology and interpretation of results.

Comments from the Academic Editor

1. Style requirements

Please ensure that your manuscript meets PLOS ONE's style requirements, including those for file naming. The PLOS ONE style templates can be found at https://journals.plos.org/plosone/s/file?id=wjVg/PLOSOne_formatting_sample_main_body.pdf and https://journals.plos.org/plosone/s/file?id=ba62/PLOSOne_formatting_sample_title_authors_affiliations.pdf

The headings have been set in bold and size 18. Figures are presented in separate files, named as requested, with titles in bold, and are cited in the text according to the journal’s formatting guidelines. The headings have been revised to comply with the required levels. Tables have been inserted into the main text, and the data and results have been right aligned for improved readability.

2. Data availability statement

In the online submission form, you indicated that “Summary and anonymized data would be available on the basis of a written reasonable request”. All PLOS journals now require all data underlying the findings described in their manuscript to be freely available to other researchers, either: 1. In a public repository, 2. Within the manuscript itself, or 3. Uploaded as supplementary information. This policy applies to all data except where public deposition would breach compliance with the protocol approved by your research ethics board. If your data cannot be made publicly available for ethical or legal reasons (e.g., public availability would compromise patient privacy), please explain your reasons on resubmission and your exemption request will be escalated for approval.

In accordance with PLOS ONE’s data availability policy, we have modified the Data Availability Statement to indicate that all data underlying the findings will be provided as supplementary information files. The sentence has been modified in the manuscript to read as follows:

“All data underlying the findings described in this manuscript are provided as supplementary information files.”

3. Ethics statement location

Your ethics statement should only appear in the Methods section of your manuscript. If your ethics statement is written in any section besides the Methods, please move it to the Methods section and delete it from any other section. Please ensure that your ethics statement is included in your manuscript, as the ethics statement entered the online submission form will not be published alongside your manuscript.

The ethics statement has been moved to the Methods section, as requested. It now reads as follows:

“Ethical approval and consent: This study was approved by the Ethics Committee ‘Nord-Ouest III’ (France) under protocol number 17-HPNCL-03. Written informed consent was obtained from all participants and/or their legal guardians prior to inclusion in the study.”

4. Tables format

Please include your tables as part of your main manuscript and remove the individual files. Please note that supplementary tables should remain/be uploaded as separate 'supporting information' files.

All tables have been inserted into the main manuscript, as requested, and the individual table files have been removed. We do not have any supplementary tables to include as “supporting information” files.

5. Reviewer citation suggestions

Acknowledged, thank you very much.

Reviewer #1

1. Inclusion of tables in the main manuscript

To begin with, although the results are promising, I noticed that the tables were not included in the PDF file. This made it somewhat difficult to accurately interpret the findings. If possible, I would appreciate the opportunity to review the full version with the tables included.

We thank the reviewer for pointing this out. Indeed, the tables were not included in the main manuscript but were provided in a separate file. We have now inserted all tables into the main manuscript, in accordance with the journal’s author guidelines.

2. Terminology for AVH

I am curious as to why the term “acoustic-verbal hallucinations (AVH)” was chosen instead of directly referring to “auditory hallucinations with voice commanding.”

We thank the reviewer for this question. We chose the term “acoustic-verbal hallucinations” (AVH) because it is broader and more frequently used in the literature to describe the phenomenology of hallucinations in children and adolescents with PTSD. In this population, reported experiences are not limited to imperative or commanding voices. Patients may also describe hearing a voice calling them by their first name, speaking to them in a neutral tone, or voices talking to each other, without giving orders. Using the broader term AVH therefore better reflects the clinical diversity observed in our sample and aligns with current research terminology (e.g., Waters et al., 2012; McCarthy-Jones et al., 2014; Jardri et al., 2014).

- Waters, F., et al. (2012). Auditory hallucinations in schizophrenia and nonschizophrenia populations: A review and integrated model of cognitive mechanisms. Schizophrenia Bulletin, 38(4), 683–693.

- McCarthy-Jones, S., et al. (2014). Hallucinations in children and adolescents: An updated review and practical recommendations for clinicians. Schizophrenia Bulletin, 40(Suppl_4), S703–S715.

- Jardri, R., et al. (2014). The neurodynamic organization of modality-dependent hallucinations. Cerebral Cortex, 23(5), 1108–1117.

3. Baseline differences

Regarding the statement that “no difference was found in emotional and social cognition at baseline,” does this refer to no between-group differences? Since the mean scale scores were not reported, I wonder whether both groups (being PTSD patients) showed similarly impaired emotional and social cognition at baseline; and perhaps only after six months did group differences emerge due to improvement in some individuals?

We thank the reviewer for this request for clarification. Yes, at baseline, social cognition scores measured with the NEPSY-II (Theory of Mind and Affect Recognition subtests) and emotional cognition scores measured with the Differential Emotions Scale were equivalent between the “case” and “control” groups. We have clarified this point in the manuscript to avoid any ambiguity with the following sentence:

“At T0, social cognition scores measured with the NEPSY-II (Theory of Mind and Emotion Recognition subtests) and emotional cognition scores measured with the DES-IV did not differ significantly between the ‘case’ and ‘control’ groups.”

Furthermore, as suggested, we have specified in the Results section the between-group comparison at baseline for the variables of interest by adding the following sentence:

“Participants with AVH at T1 had at T0, significantly higher malevolence scores in voices (BAVQ-R) and guilt levels (DES IV) compared with those without AVH at T1 (malevolence: p = 0.04; guilt: p < 0.05). Both malevolence and guilt scores at T0 were also significantly correlated with the presence of AVH at T1 (malevolence: r = 0.36, p = 0.04; guilt: r = 0.35, p < 0.05).”

4. Definition of peer bullying

Given that peer bullying constituted a relatively high proportion of trauma types in the sample, I would appreciate clarification on how "peer bullying" was defined in this study.

We thank the reviewer for this request for clarification. We have modified the corresponding sentence in the manuscript to indicate that the types of trauma were categorized based on the participant’s self-report during the clinical interview. The revised sentence now reads:

“Fig 2 presents the types of trauma experienced by the participants, based on their self-report during the clinical interview (Fig 2).”

In our study, “peer bullying” refers to any situation in which the participant reported feeling threatened or intimidated by others (children, adolescents, or adults) in their school or social environment. This definition is consistent with the commonly accepted description of bullying as intentional, repeated aggressive behavior occurring in a context of power imbalance (Olweus, 1993).

5. Typographical error in Figure 3

In Figure 3, there appears to be a typographical error at T0: the label “control group” should be “AVH–.”

We thank the reviewer for noticing this. The label has been corrected accordingly.

Reviewer #2

1. Study objectives wording

The current wording “The primary objective of our study” and “The secondary objectives of our study” may imply that the secondary objectives are unrelated. However, the listed secondary objectives appear to elaborate on and further specify the primary objective. Therefore, I suggest revising “The secondary objectives of our study are” to “Specifically, we aim to” or “To this end, we further aim to”, which would better reflect their relationship.

We thank the reviewer for this helpful suggestion regarding the formulation of our study objectives. We have revised the paragraph to ensure that the secondary objectives are clearly presented as an elaboration of the primary objective. The paragraph in the manuscript now reads as follows:

“The primary objective of our study is to compare emotional and social cognition markers in a population of children and adolescents with PTSD, between those presenting with AVH and those without AVH. Specifically, we aim to:

1. Compare emotional and social cognition markers in a population of children and adolescents with PTSD, between those presenting with persistent auditory AVH at 6-month follow-up and those without AVH at 6 months;

2. Assess the correlations between emotional and social cognition markers and the presence of AVH at baseline and their persistence at 6-month follow-up;

3. Examine the associations between emotional and social cognition markers and the clinical evolution of PTSD between baseline and 6-month follow-up;

4. Investigate the associations between emotional and social cognition markers and the emergence of a psychotic disorder at 6-month follow-up.”

2. Trauma exposure comparison

Although all participants met the DSM-5 criteria for PTSD, was there a comparison between the AVH and non-AVH groups in terms of the nature or severity of the traumatic events they experienced? Were the stressors similar across groups, or were there meaningful differences? Additionally, did the two groups differ in the time interval between the occurrence of the stressful event and their inclusion in the study? The current mental state of children and adolescents may also be influenced by how much time has passed since the traumatic event.

We thank the reviewer for raising this important question. Indeed, in our study we did not compare the AVH+ and AVH– groups in terms of the objective nature or severity of the traumatic events they experienced. Our inclusion criterion was a DSM-5 PTSD diagnosis, emphasizing the participant’s subjective experience of trauma rather than an external judgment of one event’s severity over another. This approach aligns with contemporary conceptualizations that highlight the primacy of the individual’s appraisal and experienced impact of trauma as key determinants of PTSD development, irrespective of the objective severity of the event. For instance, recent research underscores that “by focusing on traumatic experiences rather than just traumatic events, it is possible to consider a wider range of situations that may trigger post-traumatic symptoms” (Rossi and al., 2024). Additionally, we did not collect data on the time elapsed between the traumatic event and study inclusion, as many participants reported complex or repeated trauma exposures over time, with decompensation sometimes following a later event rather than the initial one. We now acknowledge this as a limitation in the Discussion.

We have added the following paragraph to the Discussion section of the manuscript:

« This study did not include a comparison between the AVH+ and AVH– groups regarding the objective nature or severity of the traumatic events, as our inclusion criterion was based on meeting DSM-5 diagnostic criteria for PTSD, which emphasizes the subjective experience of trauma (107). In addition, no data were collected on the exact time elapsed between the traumatic event and study inclusion, particularly as many participants reported complex or repeated trauma exposures over time, and PTSD decompensation could occur following a later event rather than the initial one. These factors should be considered as limitations of the present study”.

3. Diagnostic validity and assessor qualifications

Diagnostic accuracy is critical. Who administered the K-SADS-PL and the MINI-Kid (Mini International Neuropsychiatric Interview for Children and Adolescents)? Did these assessors receive formal training or hold any relevant professional licenses or certifications?

We thank the reviewer for this question. All clinical interviews and psychometric assessments, including the K-SADS-PL and the MINI-Kid, were conducted by the same evaluator, a child and adolescent psychiatrist and PhD student. This ensured both clinical expertise in administering these instruments and consistency across all evaluations in the study. We have clarified this point in the Methods section, which now states:

“All clinical interviews and psychometric assessments, including the K-SADS-PL, MINI-Kid, WISC-IV, NEPSY-II, BAVQ-R, and DES-IV, were administered by the same evaluator, a child and adolescent psychiatrist and PhD student, ensuring both clinical expertise and consistency across all assessments.”

4. WISC-IV administration

Who administered the WISC-IV? Did the examiners receive standardized training or hold professional certification to ensure the reliability and validity of the assessment?

We thank the reviewer for this question. In our study, an abbreviated version of the WISC-IV was administered, consisting of four subtests (one per index): Similarities for the Verbal Comprehension Index, Picture Concepts for the Perceptual Reasoning Index, Digit Span for the Working Memory Index, and Symbol Search for the Processing Speed Index. The purpose of this abbreviated assessment was not to provide a precise full-scale IQ score but rather to screen for intellectual disability (Full Scale IQ < 70) as a non-inclusion criterion. This short-form WISC-IV has been validated for screening purposes, showing good sensitivity and specificity in detecting intellectual disability (e.g., [reference to add]). IQ scores were not included among the variables compared or correlated in the study objectives. All WISC-IV assessments were conducted by the same evaluator, a child and adolescent psychiatrist and PhD student, who had been specifically trained in test administration by the service’s neuropsychologists.

We have clarified this point in the Methods section, which now states:

"French version of the abbreviated form of the WISC-IV (Similarities, Picture Concepts, Digit Span, Symbol Search) (56) was administered to screen for intellectual disability (Full Scale IQ < 70) as a non-inclusion criterion; this short form has demonstrated good sensitivity and specificity for scre

---

## [Decision Letter · Decision Letter 1]

8 Sep 2025

Relationships between markers of emotional and social cognition and acoustic-verbal hallucinations in children and adolescents with post-traumatic stress disorder (PTSD)

PONE-D-25-30151R1

Dear Dr. Arnaud Fernandez,

We’re pleased to inform you that your manuscript has been judged scientifically suitable for publication and will be formally accepted for publication once it meets all outstanding technical requirements.

Kind regards,

Mu-Hong Chen, M.D., Ph.D.

Academic Editor

PLOS ONE

Additional Editor Comments (optional):

Reviewer #1:

Reviewer #2:

Reviewers' comments:

Reviewer's Responses to Questions

**Comments to the Author**

1. If the authors have adequately addressed your comments raised in a previous round of review and you feel that this manuscript is now acceptable for publication, you may indicate that here to bypass the “Comments to the Author” section, enter your conflict of interest statement in the “Confidential to Editor” section, and submit your "Accept" recommendation.

Reviewer #1: All comments have been addressed

Reviewer #2: All comments have been addressed

2. Is the manuscript technically sound, and do the data support the conclusions?

Reviewer #1: Yes

Reviewer #2: Yes

3. Has the statistical analysis been performed appropriately and rigorously? 

Reviewer #1: Yes

Reviewer #2: Yes

4. Have the authors made all data underlying the findings in their manuscript fully available?

Reviewer #1: Yes

Reviewer #2: No

5. Is the manuscript presented in an intelligible fashion and written in standard English?

Reviewer #1: Yes

Reviewer #2: Yes

6. Review Comments to the Author

Reviewer #1: Thank you for the opportunity to review this interesting manuscript. In my clinical practice, auditory verbal hallucinations are a common symptom among children and adolescents with a history of trauma. I appreciate the authors’ clear responses to the questions regarding the definition and terminology of AVH, as well as to the other points raised. I have no further comments on the current results and discussion.

Furthermore, I am particularly curious about the clinical course of individuals who transition from not experiencing AVH to developing AVH, as well as those who show the opposite trajectory (from experiencing AVH to remission). It would be valuable to know whether these changes are associated with pharmacological or other therapeutic interventions. I look forward to future results that may shed light on these important aspects.

Reviewer #2: there were not additional comments for the author, including concerns about dual publication, research ethics, or publication ethics.

7. PLOS authors have the option to publish the peer review history of their article (what does this mean? ). If published, this will include your full peer review and any attached files.

**Do you want your identity to be public for this peer review?** For information about this choice, including consent withdrawal, please see our Privacy Policy .

Reviewer #1: No

Reviewer #2: **Yes: ** H.-H. Lu

---

## [Editor Report · Acceptance letter]

PONE-D-25-30151R1

PLOS ONE

Dear Dr. Fernandez,

I'm pleased to inform you that your manuscript has been deemed suitable for publication in PLOS ONE. Congratulations! Your manuscript is now being handed over to our production team.

Kind regards,

on behalf of

Dr. Mu-Hong Chen

Academic Editor

PLOS ONE